# H^+^ and Confined Water in Gating in Many Voltage-Gated Potassium Channels: Ion/Water/Counterion/Protein Networks and Protons Added to Gate the Channel

**DOI:** 10.3390/ijms26157325

**Published:** 2025-07-29

**Authors:** Alisher M. Kariev, Michael E. Green

**Affiliations:** Department of Chemistry and Biochemistry, City College of the City University of New York, New York, NY 10031, USA; akariev@ccny.cuny.edu

**Keywords:** channel gating, confined water, protons, hydrogen bonds, ion hydration

## Abstract

The mechanism by which voltage-gated ion channels open and close has been the subject of intensive investigation for decades. For a large class of potassium channels and related sodium channels, the consensus has been that the gating current preceding the main ionic current is a large movement of positively charged segments of protein from voltage-sensing domains that are mechanically connected to the gate through linker sections of the protein, thus opening and closing the gate. We have pointed out that this mechanism is based on evidence that has alternate interpretations in which protons move. Very little literature considers the role of water and protons in gating, although water must be present, and there is evidence that protons can move in related channels. It is known that water has properties in confined spaces and at the surface of proteins different from those in bulk water. In addition, there is the possibility of quantum properties that are associated with mobile protons and the hydrogen bonds that must be present in the pore; these are likely to be of major importance in gating. In this review, we consider the evidence that indicates a central role for water and the mobility of protons, as well as alternate ways to interpret the evidence of the standard model in which a segment of protein moves. We discuss evidence that includes the importance of quantum effects and hydrogen bonding in confined spaces. K^+^ must be partially dehydrated as it passes the gate, and a possible mechanism for this is considered; added protons could prevent this mechanism from operating, thus closing the channel. The implications of certain mutations have been unclear, and we offer consistent interpretations for some that are of particular interest. Evidence for proton transport in response to voltage change includes a similarity in sequence to the H_v_1 channel; this appears to be conserved in a number of K^+^ channels. We also consider evidence for a switch in -OH side chain orientation in certain key serines and threonines.

## 1. Introduction

Ion channels, including voltage-gated ion channels, have been the subject of intensive research for the better part of a century. Their importance in a wide range of diseases, as well as their role in critical biological functions, including nerve pulses, has justified this effort. There is a range of channels; all channels must carry out a set of functions, including the following: (1) gating, i.e., the control of the channel’s opening for conduction; (2) selectivity—choosing what to conduct; and (3) ancillary functions, including ending conduction, which is often accomplished via inactivation. The latter is a state in which the channel undergoes a transition that stops conduction, prior to returning to the original closed state so that the channel can repeat the cycle. Control of gating may be effected by a change in membrane potential for voltage-gated channels, or by ligands (ligand-gated channels), or by other environmental variables (e.g., temperature and pressure). At least some channels have protons responsible for gating, or possibly partially responsible for gating. There are now known structures of channels, which began with work by MacKinnon and colleagues a quarter of a century ago [1]. Since then, there has been considerable effort devoted to understanding the mechanism by which channels operate. In this paper, we consider the gating of voltage-gated channels, with emphasis on certain potassium channels; this paper shows in detail how a proposal first made in 1989 [2] has been, in part, confirmed by quantum calculations. In that paper, one of us proposed that the gate consisted of water, which was frozen (immobile) when closed and essentially liquid when the channel was open. The proposed mechanism by which this, and the gating current, occurred has long since been superseded—in fact, describing the water as “frozen” can no longer be considered correct, although at the present moment, the fundamental idea that the arrangement of water is crucial in gating still appears to be correct. In this review, we refer to our earlier work showing that the gating current may involve protons [3,4,5,6], along with evidence in the rest of the literature that is consistent with this mechanism. Most of the evidence has been interpreted by the majority of workers in the field in terms of a mechanical gating model. Figure 1, with no water, illustrates the traditional view of the channel. We consider reasons for interpreting the data in terms of protons as the gating current.

To understand the gate and the effect of the changes in water structure as affected by protons, we can draw on a huge volume of research on water in boundary layers, especially in the boundary layers of proteins. In addition, we can see that the dimensions of the gate are of the order of a few Angstroms to perhaps two nanometers, with a thickness of the gating region also on the order of nanometers. This means the water is confined water, which is known to be different in important properties from bulk water.

A gating mechanism proposed here involves water and proton motion, but not S4 motion: the generally accepted gating model has a segment, S4, for each of the four voltage-sensing domains (VSD) moving in response to an external electric field, which compresses the gating region when the voltage is on, thus closing the channel. Each VSD has four transmembrane helices; S4 has a series of positive charges, mainly arginines, and is considered to be the helix that moves in response to an external field. When the voltage is removed, S4 is believed, in this class of models, to return to its position within the membrane next to the negatively charged S2 and S3 helices. There is also considerable evidence that is difficult to interpret in a gating mechanism that involves motion of S4; this evidence is often ignored. We consider the question of the structure of the closed state, but the open state structure has been illuminated in high resolution using X-ray crystallography, and recently using cryoEM. If, as we postulate, protons enter the gating region to close the channel, this must work by rearranging the water and affecting the hydration state of the ion. This is all conditional on the boundary conditions provided by the protein; the boundary can also rearrange (e.g., a serine or threonine hydroxyl may reorient, or an aspartate may be neutralized with further effects as a consequence) in response to protons entering the gating region. The dimensions of this region are less than 2 nm, so protons can also change the hydration state of the permeant ion. In the region, there are, in addition to the permeant ions, protein side chains, water, counterions, and—depending on the state of the channel gating cycle—additional protons. Together, these form structures that either allow the permeant ion to move forward or block it. The only protein motion we have in the model we propose is the motion of side chains, while the motion of the protein backbone, if any, is very small and not relevant to gating. The subject has implications that extend to the behavior of small cavities that are ubiquitous in biological cells.

## 2. Major Topics

(1)The H_v_1 channel as an analogue of the VSD.(2)There is evidence from sodium channels that gating is slowed by D_2_O.(3)Certain mutations in the gate region make a huge difference.(4)Proton concentrations at the gate can be very high.(5)Wallace and coworkers found, in a bacterial sodium channel, Na_v_Ab, that hinge motions of the pore helix S6 and the C-terminal domain were involved with gating, but not S4 motion.(6)T1, the intracellular segment of the channel protein, matters.(7)The piquito, a brief pulse at the start of the channel opening sequence—proton tunneling?(8)Much of the evidence for the standard mechanical model is derived from experiments in which arginine was swapped for cysteine.(9)Other residues could be mutated as well.(10)The interactions that exist in the gate include ion–ion and hydrogen bonds.(11)Many molecular dynamics calculations on ion channels have been published.(12)Confined water, especially in channels.(13)Hydration dynamics.(14)In addition to the prolines at the gate, there is another class of conserved residues.(15)The EAG channels are also potassium channels.(16)Vibrations and rotations.(17)Thermodynamic behavior.(18)The probable importance of protons.(19)Experiments that appear to be more consistent with the standard model.(20)Energetics.(21)Omissions from this review.

We can see how the evidence points to a gating mechanism in which the gating current consists of proton motion, not motion of the S4 segment of the VSD. The structure of the water as modified by ions in the confined space defined by the structure of the side chains of the boundary proteins determines the gating mechanism.

Review of the evidence: We list the main sets of evidence related to the questions we consider, organized into topics that cover the most important aspects of each question.

### 2.1. The H_v_1 Channel as an Analogue of the VSD

There are channels known to conduct protons. The H_v_1 channel has a structure that is, in many ways, similar to the VSD [7,8,9,10], although there are significant differences (for one thing, H_v_1 is normally a dimer, although it can function as a monomer), and the proton path would differ somewhat from that in the VSD of the mammalian K_v_1 channel; the upper (extracellular) part of the channel strongly resembles that of the VSD of the K_v_1.2 channel, while the path differs somewhat in the intracellular section, with H_v_1 going directly to the intracellular side of the membrane, whereas K_v_1.2 deviates toward the pore). Figure 2 illustrates a putative proton path. Although it is not generally considered that this challenges the standard model, it is difficult to see why it does not; this provides a similar path that could be taken by protons [11]. H_v_1 has been the subject of multiple studies, and there is still no general agreement on its gating mechanism [8,12,13,14,15,16,17]. Of particular interest, it appears that protons contribute to gating current, and there is pH sensitivity as well as voltage sensitivity in gating [8,14,18,19,20]. While there are a variety of models in which the S4 segment moves, the fact that a proton gradient can at least contribute to gating suggests that voltage gating may work via a proton cascade. Protonation of an aspartate on the S1 segment of K_v_1.2 as part of the mechanism of proton transport [20] would also suggest that proton transport may play a role in voltage gating, although admittedly this is indirect evidence. A number of residues, when mutated, change the gating, transport, and other properties of the channel [7,8,21,22], and some lead to a contribution of protons to the gating current. Channels with a rather different structure also transmit protons and suggest alternate proton-transmitting paths, such as in the M2 channel of the influenza virus; specific paths for the A and B variants have different paths, and there are different pKa values for specific histidine residues [23]. Some of the steps in this channel are almost certainly applicable to other ion channels, including those considered here. However, this path differs from the voltage-gated potassium channel by a significantly greater amount than H_v_1.

### 2.2. There Is Evidence from Sodium Channels That Gating Was Slowed by D_2_O [24,25,26]

Furthermore, different effects of D_2_O are found on different phases of gating. Starkus, Rayner, and coworkers found at least two stages in gating that were affected by D_2_O substitution; in particular, this included the finding by Alicata et al. that the last step in gating a sodium channel was slowed by D_2_O [27,28,29]. Standard models that ignore water other than as a background medium cannot account for this. Molecular Dynamics (MD) simulations are part of the evidence used to support standard models. MD more often than not uses TIP3P water, which is not a good model for water, particularly for confined water. Because D_2_O is important, an accurate water model is needed, including, at a minimum, polarization [30]. If D_2_O makes a difference, then water must actually be part of the gating mechanism. If protons are important, this is expected. On standard models, it is hard to understand why D_2_O matters. D_2_O has also been found to affect inactivation by changing the nature of the hydration around the pore, rather than within the pore [31]. The possible non-specific effects of D_2_O, such as viscosity, were ruled out. Given that there is coupling between inactivation and the gate at the entrance to the pore [32], this may be further evidence that structural water is part of the gating mechanism; this is the most economical interpretation of the data.

### 2.3. Certain Mutations in the Gate Region Make a Huge Difference

Particularly important are proline mutations in the highly conserved PVPV sequence at the gate. This includes, in particular, aspartate and serine mutations (Figure 3). Aspartate mutation produces a channel essentially constitutively open at physiological potentials [33,34,35]. This is qualitatively relatively easily understood on the proton model, but not on the mechanical model. On the proton model, aspartate can be protonated, making the carboxyl group neutral, thus removing the protons from the gate and allowing it to operate like an open, unprotonated gate. Substitution of serine led to a channel that was closed at all physiological potentials. The serine side chain is smaller than the aspartate (or proline) side chain, meaning that the channel should have its gate wide open if the mechanical model were correct. The fact that the serine mutant is closed does not seem consistent with this model, but makes perfectly good sense if potassium transit through the pore, and especially near the proline level of the gate, requires dehydration of the K^+^ ion. The potassium must be at least mostly dehydrated; if it reached the cavity at the central portion of the pore with all six hydrating water molecules, then there would have to be a countercurrent of water that would block the passage of following ions, almost certainly for long enough to limit the current severely. Dehydrating the ion requires that the gate opening be correctly spaced to allow the local amino acids to strip the water from the potassium ion—too wide an opening would prevent that. This, of course, requires an answer to why the gate works at all, if the spacing is important but is the same in open and closed states (see Section 2.15 below). Protons at the gate could have several effects, including rotating the side chains of the gate amino acids and forming networks that include water, side chains, and the ions, including counterions. Side chains in one orientation could dehydrate the potassium ion, while in the other conformation, they fail to do so, and the ion is unable to progress into the pore cavity—the difference between open and closed would thus not require motion of the protein backbone but would be a local effect of the protons on the side chain orientation. Serine is too small to bring about dehydration—a wide-open gate does not allow the ion to move forward from the gate into the cavity.

### 2.4. Proton Concentration at the Gate Can Be Very High

In a 1 M H^+^ solution (pH 0, if one ignores activity coefficients), there are about 55 water molecules for each proton (half that, if the other half are hydrating the counterion). With four protons in a region containing perhaps 20 water molecules, meaning the equivalent of somewhere between 5 and 10 M, the effective concentration is much greater. There are lots of uncertainties in detail—counter ions, the number of water molecules, and activity coefficients, to begin with. It is safe to say the region is as strongly acidic as is possible, so molecules that are strong acids in bulk can still allow protonation, or at least ion pairing, under the circumstances present at the gate. It is not silly to postulate protonated amides, for example. H^+^–Cl^−^ ion pairs are likely, although HCl is a strong acid in bulk. The actual distribution of the protons cannot be guessed, but must be determined by calculation; the protons behave as quantum particles, and should be calculated as such. Unfortunately, standard quantum calculation software depends on the Born–Oppenheimer (B-O) approximation, even for protons, so the protons are not treated completely correctly; however, the remainder of the system and the molecules are treated reasonably accurately, and we must assume the B-O approximation does not introduce so much error as to invalidate the calculation. If there are no protons, there is still a concentrated solution of ions and side chains.

### 2.5. Wallace and Coworkers Found, in a Bacterial Sodium Channel, Na_v_Ab, That Hinge Motions of the Pore Helix S6 and the C-Terminal Domain Were Involved with Gating, but Not S4 Motion

Wallace and coworkers found, in a bacterial sodium channel, Na_v_Ab, that hinge motions of the pore helix S6 and the C-terminal domain were involved with gating, and affected gate diameter, but there was essentially no S4 movement between open and closed structures [37,38]. It is conceivable that for some reason, not at all clear, this channel is unique; although it has some differences with other channels, it has the same fundamental structure as all other members of the superfamily that includes both it and the type of channel we consider here. This is strong evidence that S4 does not need to move to effect gating. However, something must move in response to voltage without changing the structure of the channel. Protons are one thing that could fulfill this requirement [6].

### 2.6. T1, the Intracellular Segment of the Channel Protein, Matters

T1, the intracellular segment of the channel protein, matters [39]—in standard models, why? No version of a standard model seems to accommodate this; T1 seems to be in the wrong place for a van der Waals push between the S4 and the gate—it stretches “down” (i.e., intracellular direction) and pulls the linker away from the gate if pushed down by S4. We found a proton path that extends through T1, between S4 and the gate [6]. There is also a paper by Cushman et al. that reinforces this point; substituting aspartate for asparagine in T1 near the T1 to gate junction leads to left-shifted gating and larger current [40]. Proton transfer is one possibility that would account for this experimental finding. The carboxyl group is known to be efficient in dehydrating potassium; we also observed that the distance of T1 from the membrane surface is less than 2 nm, and thus contains confined water. Aspects of the space separating the membrane, or its surface, from T1, may affect the transport of anything, but especially protons, from the VSD to the pore.

### 2.7. The Piquito, a Brief Pulse at the Start of the Channel Opening Sequence—Proton Tunneling?

There is a brief pulse at the beginning of the gating current that has been labeled the “piquito” (a slightly slower pulse is definitely associated with the gating current) [41,42]; it has fairly well defined characteristics: timing <2 μs (how much less is unknown), or 12 μs for the slower example, and magnitude (around 1% of gating current). This can be accounted for by proton tunneling, which can move a charge perhaps 1 Å. If this is one of three protons to move, it would be about 1/3 of the charge associated with one VSD. The observed charge motion, about 1% of gating current, would be about right if it moves 1/3 of the total charge (i.e., about one charge) through about 1/30 of the distance, which, for a 1 Å distance makes sense, more or less—this is too rough a “calculation” to expect much better accuracy, but at least the order of magnitude is right). It is possible to estimate the transmission probability for a proton through a 1 to 2 Å barrier with and without a field applied using the WKB approximation [43]. For a barrier about 20 k_B_T high, it is possible to shape the barrier to obtain a result that agrees with the tunneling hypothesis. However, absent any specific evidence for a particular barrier shape, the most that can be said now is that the hypothesis is plausible and can agree with the observation. If one proton begins to transfer, it would become the start of a very limited cascade of probably three protons; transferring these through the voltage drop would produce the measured gating current. A mechanical model might suggest perhaps an arginine side chain flip as the piquito, but why would this be coupled to the motion of the entire S4 segment? A more plausible side chain flip that need not be associated with proton tunneling would be a reorientation of the side chain of a phenylalanine that could be associated with the transient proton in the VSD. Although the nature of this association is not certain to be direct, it would be a H^+^–π electron association. There is a phenylalanine, F222, in the VSD of the 3Lut (pdb) [44] structure that appears to be important for transition to the open structure of the K_v_1.2 potassium channel. F222 is located not far from the arginine at which the large voltage drop found by Asamoah et al. [45] occurs. An F222 side chain flip would still accompany a proton gating model, as it is not associated with an actual translation of S4, but might assist in transferring a proton. The piquito makes perfect sense as part of a proton gating model, both in terms of the order of magnitude that might be expected and the effect of a localized field like that reported by Asamoah et al. Further work would be required to move this hypothesis to a more certain footing; so far, it can be said that proton tunneling can be a plausible mechanism, and it is very difficult to reconcile this result with a mechanical model.

### 2.8. Much of the Evidence for the Standard Mechanical Model Is Derived from Experiments in Which Arginine Was Swapped for Cysteine

Much of the evidence for the standard mechanical model is derived from experiments in which arginine was swapped for cysteine; the cysteine was then reacted with an MTS (methyl thiosulfonate) reagent, a technique developed by Yang and Horn [46]. Originally, it was assumed that cysteine, which must ionize to react, must move to the membrane surface to ionize, to get out of the low dielectric environment of the membrane. Actually, the cysteine side chain is tiny compared to arginine, so water, as well as MTS reagents, can reach the cysteine in situ. The existence of aqueous crevices in the VSD was discovered not long after Yang and Horn’s original publication. F222, as discussed above, appears to be the only hydrophobic barrier between the intracellular and extracellular crevices. Cysteine can therefore presumably ionize without moving. Replacing arginine with cysteine makes even more space available. However, differences in the reaction of different substituted cys with MTS reagent between open and closed states of the channel were observed. The apparent accessibility of the cysteines was interpreted as demonstrating the motion of the cysteines, and thus S4. As we do not have the crystal structure of the mutated VSD, it is not appropriate to make a quantitative assertion as to the dimensions of the available space. We note, however, that the guanidinium group of arg is approximately the size of a methyl thiosulfonate group, while the cysteine side chain, consisting of a single atom when ionized (not much bigger with the H atom added when not ionized), takes up almost no space. If the R→C mutation opens more space for an MTS reagent, the primary evidence for S4 motion can be accounted for without S4 motion. By now, the existence of an aqueous crevice through most of the VSD is understood, with only a narrow hydrophobic barrier between internal and external aqueous accessibility, as demonstrated by Starace and Bezanilla [47] and suggested even earlier by Yang and Horn [48]. With an R→H mutation, domains I, II, and III of a standard sodium voltage-gated channel enable a proton current, as the histidine allows enough proton passage to provide a measurable continuous current. The aqueous crevices in these three domains are comparable to those in K_v_ channels like Shaker [49]. This does not make it certain that the three protons needed for gating current could go through the native VSD, but it does make it plausible that a few protons could go through with a change in potential.

### 2.9. Other Residues Could Be Mutated as Well

For example, Horn and Nguyen mutated residues on S3, which have a negative charge. In the simplest interpretation, this should have shown opposite accessibility compared to states from the S4 results, but it showed the same accessibility [50] and thus the same apparent motion as S4. In spite of this, the result was reinterpreted as consistent with the standard model. A paper by Naranjo and coworkers found that substituting aspartate for arginine in the VSD, again reversing charge, had little effect on the gating current [51]. It did not reverse the direction of the gating current, as might have been expected from the charge reversal if S4 were mobile, although it produced other effects, particularly a large volume change near S2 and S3. The latter result is much more easily understood with proton gating, in which the cysteine accessibility did not depend on physical movement of the cysteine. The authors, however, maintained an interpretation that is consistent with the standard mechanical model. In a different section of the channel, Bezanilla and coworkers mutated the extracellular linker S3-S4, truncating the linker down to 0, 5, or 10 residues [52]. Presumably, with 0 residues, there should be no motion of the S4 (unless it dragged the S3 with it, a very energetically expensive transition). The other truncations, especially to five amino acids, should have limited the motion greatly. In fact, all three mutations produced functioning channels. The zero residue case had a smaller gating current and slowed the channel appreciably, but still produced a functioning channel. The other mutations only slowed the opening. It is difficult to see how this is consistent with a large movement of S4.

### 2.10. The Interactions That Exist in the Gate Include Ion–Ion and Hydrogen Bonds

Hydrogen bonds are often considered largely electrostatic, but they need not be; there are hydrogen bonds that depend on partially covalent bonds [53,54,55,56]. Even ordinary water has hydrogen bonds with a partially covalent character [57]; the fact that hydrogen bonds may be anti-electrostatic, or at least partially covalent, tells us to be careful about interpreting hydrogen bonds too simply—that is, in purely electrostatic terms. Ions have additional effects on the water. They may interact in ways that are suggested by the Hofmeister series [58]. Guanidinium, the side chain of arginine, an amino acid that is critical to gating, is well known to disrupt water’s structure; it also interrupts hydrogen bonding [59]. Two- and three-body interactions can produce clusters of multiple water molecules [60]. These effects can be expected in the limited space of the gating region, with the side chains of the amino acids.

### 2.11. Many Molecular Dynamics Calculations on Ion Channels Have Been Published

Many molecular dynamics calculations on ion channels have been published. These usually claim to show how some version of the standard model is correct, and have been recently reviewed by Phan et al. [30]. There are certain problems with these calculations: (i) These almost always use fields an order of magnitude too large, and distributed over the entire VSD. However, the voltage drop occurs only over a very limited region, essentially one arginine [45]; the field at this one arginine is at least 10^8^ V m^−1^, similar to that used by the MD simulations, but it must be very local. The equation 10^8^ V m^−1^ × 7 × 10^−10^ m = 70 mV indicates the total available voltage. If one uses the field across the entire 70 × 10^−10^ m thickness of the membrane, one needs 700 mV, which is an order of magnitude out of the physiological range. In addition, Onsager showed that fields this large produce an increased probability of ionization of weak electrolytes via the second Wien effect [61]; since Onsager’s calculation depends on the local dielectric constant, it is difficult to apply quantitatively to the channel, but the effect must be appreciable. To produce a proton cascade, a weak electrolyte must ionize; if the MD calculations could take this into account, perhaps they too would find a proton cascade. By the time one gets to the fields used in MD simulations, this field may cause ionization through the entire VSD, although this would not appear in a classical calculation. The fields of 10^8^ V m^−1^ are large enough in any case to create a non-physical situation, except for the one arginine, where the entire field drops [45]. If the local field at the one arginine where the field is this large does not produce ionization by the second Wien effect, it may also lower the barrier for proton transfer, allowing tunneling to occur—a completely different mechanism, but one that also requires the high field to be very local; in either case, 70 mV is all the voltage that there is. (ii) The water model most frequently used, TIP3P, does not enable a good representation of the water in a confined space. In particular, this must be a problem at the gate of the channel. This suggests that steps involving water in gating will be given erroneous configurations in MD simulations, which may be critical in itself. In addition, the protonation state cannot change during a classical simulation, so in many simulations, the protonation state is assumed to be that in bulk solution at pH 7, not necessarily the correct set of protonation states; the incorrect protonation state can produce seriously incorrect results [62]. Even better models than TIP3P, if they are not polarizable, do not make a serious advance in accuracy; polarizable potentials are very expensive, and therefore uncommon. It is not only polarizability that is a problem. The underlying force field models are also uncertain. For example, a study of arginine–phosphate interaction compared experimental to simulated thermodynamic values for the interaction. The authors solved the problem by adjusting a Lennard–Jones parameter in the CHARMM force field [63]. If there is no experimental result for comparison, no such adjustment is possible. While polarizable potentials are beginning to appear, they do not allow for correction of the other problems listed here, including ionization states and the inability to include ionization. Phan et al. showed that polarization may be critical in understanding the behavior of water in a channel-sized pore [30]. The behavior of water in nanopores has been thoroughly reviewed by Lynch et al. They concluded from the sum of MD studies that water is critical for transport, including proton transport. With a view to designing nanopores that can perform various functions, they observed that such a design requires accounting for the role of water [64]. Furthermore, the differences in hydrogen bonds discussed in the previous section, including the partially covalent character of some bonds, must be taken into account. There are even anti-electrostatic characteristics of some of these bonds that can only be included with a quantum calculation. This may be especially relevant in a confined space, where the exchange of water molecules hydrating ions is likely to be much slower. (iii) No MD calculation has been shown to be reversible when the voltage is removed. Over the past few years, simulations long enough to allow this to be seen have become possible, but reversibility is not seen even in new simulations. Considering that the channel protein must get through thousands of cycles before the protein is replaced, this is a serious problem. MD simulations cannot reproduce reversibility (at least, no examples seem to be in the literature, except perhaps for a case in which the final structure is targeted by steered MD). While it is not the case that all MD calculations concerning ion channel gating are necessarily incorrect, the majority of these calculations require extremely careful scrutiny when applied to ion channels, including, in particular, gating of the channels. These calculations must not include implicit assumptions that are not valid, but this has been common; however, the situation may improve as computers become capable of running simulations that include quantum effects. At least polarization and correct ionization states are needed. MD calculations that allow ionization, that is, ab initio MD, exist [65,66], but cannot be used for anything as large as an ion channel—at least, they have not been until now. Until normal electric fields and reversibility of the simulation are possible, the MD results cannot be considered as providing good evidence of the behavior of channels. Newer forms of simulation are becoming available, of which the most important is probably AlphaFold. The use of AlphaFold is problematic in this context as well, since it rarely, if ever, is trained on structures that include explicit water. So far, it appears that AlphaFold is not quite ready to use on channels. There are a few path integral or ab initio simulations, but these do not seem to include enough of the channel, at least up to this time, for them to produce gating models. When computer capabilities are sufficient, MD will become an extremely powerful tool.

### 2.12. Confined Water, Especially in Channels

In the Introduction, we noted the vast literature on the state of water in confined spaces. Much of it concerns water in nanotubes [67,68], but there is also work on hydrophobic spaces, as well as on a few channels, albeit not the channels with which we are primarily concerned here [69,70,71]. Water in nanotubes has different transport properties, which apparently include quantum coherence and proton delocalization [72,73,74,75,76,77,78]. More to the point, in the context of channels, are the many studies of proteins with water, mainly internal and bound water in the proteins. We can only list a tiny fraction of this literature here [79,80,81,82,83,84,85,86], but the gist of the results is that water with limited freedom of motion is bound in some proteins, often in internal positions in the protein. The entire subject of confined water at protein surfaces was reviewed by Bellisent-Funel and coworkers and by Laage et al. [87], among many others. Mondal et al., in a brief review, argued for the importance of water-hydrating proteins in protein function [88]. Water crowding at the protein surface, with fluctuations, is also important [89]. Spectroscopic techniques have been used to determine the rates of relaxation or transitions among boundary water molecules [90,91,92]. The distribution of relaxation times also differs at the surface [93,94]. There have been many simulations (e.g., [95]), although, as we discussed in the previous section, there is reason to be extremely cautious about accepting the results of classical molecular dynamics (MD) for confined water. Its properties often, probably generally, require the use of quantum calculations. There has been an ongoing controversy over the importance of quantum effects [76,96,97,98], but there seems to be good reason by now to take these effects into account. Obviously, the discussion in this paragraph is no more than an introduction to the fact that water at protein surfaces plays a critical role in understanding the functioning of the protein, and that that water is unlike bulk water in several ways, including its hydrogen bonding, its relaxation, and its density, among other properties. We will encounter other examples in discussion of particular characteristics of water in relation to channel gating. In channels, interactions of water with ions, forming a network, are particularly relevant, and in at least one case, a much more connected network has been found [99]. Surface layer properties almost certainly apply to membrane surfaces as well; standard models of channel gating would have parts of S4 move along the membrane surface, and proton gating models would have protons transferring across the membrane surface, so membrane surfaces are relevant to the question of channel gating on either model. Surface layers have been studied and found to have properties different from bulk water [100]. Fluctuations in water in the channel have been proposed as involved in channel properties [15]. The subject of water in channels has been reviewed, albeit largely from the point of view of simulations, but questions remain open [64]. Moving protein, here S4, along the membrane surface would also be affected by these layers. The density of water in confined spaces can approximate that of water in the high-density amorphous (HDA) phase of water that is known at low temperatures, a phase very different from room temperature bulk water [101], and this probably extends to a high water density for about two hydration shells, a distance of about 1 nm from the protein surface. This is enough to include the entire gating volume, as the gate diameter is never more than 2 nm. The ion usually enters the gate with six water molecules in its hydration shell, but it exits the channel with two at most, based on streaming current studies [102]; no model of the selectivity filter allows for more than two water molecules to leave for each potassium. Therefore, the gate must strip at least four water molecules from the potassium hydration shell when it is open. If it cannot do this, the gate is closed. In addition, the vibrations and rotations of water in a confined space are different from those in bulk. We discuss this in Section 2.16 below.

### 2.13. Hydration Dynamics

When hydration dynamics are considered, as in the fluorescence studies performed by Raghuraman and coworkers, they show considerable importance for several channel states and transitions [103]. In this case, an inverse rectifier, KirBac 1.1, was studied, and the slide segment moved less than 3 Å, which similarly limits the S4 motion.

### 2.14. In Addition to the Prolines at the Gate, There Is Another Class of Conserved Residues

In addition to the prolines at the gate, there is another class of conserved residues found in most of the voltage-gated potassium channels. These are residues that can transmit a proton. Not all the residues are the same, as some cases have asparagine, some glutamine, and some arginine or lysine. Table 1 gives some examples; note that there are essentially no hydrophobic amino acids, but there are acids, arginine, amides, and alcohols (both S and T, as well as Y).

In examining Figure 4 (below), we observe several instances where water would fit almost exactly. The diagonal of 13 Å would fit two water molecules plus one ion. The geometry can be most easily understood as forming a framework for water molecules, and holding the water that enters the gating region as the ion hydration water, but is removed. The other one or two molecules can enter the cavity, with the others stripped off. The stable configuration appears to have one water on each side of the ion next to the protein side chain. Dehydration alone is not enough; there are multiple species in the gate: K^+^, Cl^−^, H_2_O, protein side chains, and, if this review is correct, H_3_O^+^ in the closed state. This will necessarily produce a complex network in which there are several types of hydrogen bonds; water–water is only one [104]. There are hydrogen bonds of water to proteins, and electrostatic interactions of the molecules with the ions, in addition to those among the ions. The differences in the network in the open and closed states produce gating. One thing that is certain, regardless of the gating mechanism, is that partial dehydration must occur, because the ion arrives with six (usually) waters of hydration [105] and goes through the pore with two at most. There is evidence strongly suggesting dehydration for ions passing through ion channels [102,106]. It is not clear how a protein backbone motion could close the channel without major structural disruption, such that it would be difficult to return to this open structure; no apparent side chain motion appears possible that would mechanically close the channel, which would require squeezing out the water while breaking multiple hydrogen bonds, and at least disrupting multiple van der Waals interactions, at a significant energy cost. The VSD volume change would disrupt the lipid layer. The open channel has a current voltage curve that looks like a standard rectifier with current in the positive voltage region proportional to voltage. Wang and Sun showed, using quantum calculations, how dehydration is driven by the electric field, which should lead to exactly what is found [107].

Mechanical interpretations of the amino acid arrangement near the linker–pore junction have been attempted. One of the most interesting is the observation, by Chowdhury et al. [108], that there is an R–E–Y triad at that location, which they interpret as a hinge that somehow transfers a force from the linker to the pore, opening or closing it. We interpreted the same triad, which is also present in the VSD [4], as a proton transmitter. If that triad acts as a hinge near the pore, it is hard to see what it is doing in the VSD, where it cannot act as a hinge. However, if proton transport is the function of the triad, then it is consistent with functioning both as a proton transmitter near the gate and as a proton source in the VSD. Barros et al. suggested that there must be a long-range allosteric connection between VSD and gate [109]. Protons could make such a connection—if they are transported from one location to another, they would appear to have the same effect as a mechanical allosteric connection.
Figure 4Channel K_v_2.1: two partial views. (**A**) The gating region extends beyond the prolines (406); the complete gate at least includes the asparagine (410)–serine (413)–asparagine (412) intracellular to the prolines. The net structure is approximately the frustum of a cone, contracting going up the pore toward the cavity. This may help in dehydrating the ion, as the side chains compete for the water. (**B**) A view looking from the intracellular side into the pore, in which the amino acids intracellular to the prolines—asp-ser-asp—are projected onto a plane. They form an essentially octagonal figure, which looks square in this projection. The diagonal distance shown is 13.14 Å. From pdb 8SD3 [110]. The amino acid numbers are from this pdb structure.
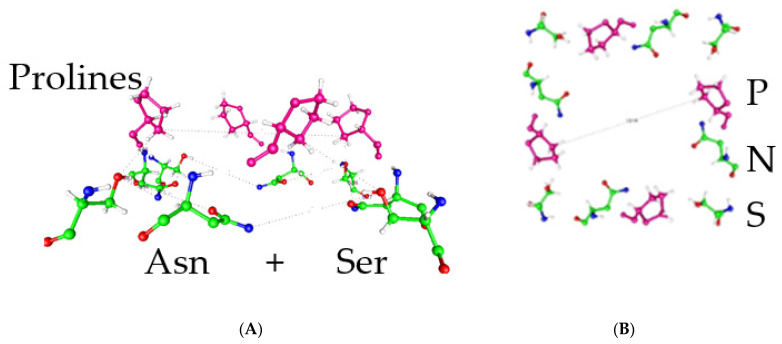


### 2.15. The EAG Channels Are Also Potassium Channels

The EAG channels are also potassium channels, albeit not domain-swapped. They have the normal set of VSDs. Closed and open states have been determined for eag2, and, in recent work by Zhang et al., a closed state exists for which there is no movement of S4 [111], with two key structures shown in Figure 5. If S4 does not move, something has to account for the apparent physical closing of the pore, as well as for the gating currents. The narrowest passage near the gate has open and closed states with almost identical diameters, as determined by interatomic distances rather than the Hole algorithm, about 4.2 Å. The authors of this eag2 study did not consider the possibility that protons could have been responsible for the rotations of amino acid side chains that they observe. We could include, as part of the gate, the side chains above the narrowest point, from tyrosine and phenylalanine. Interpreting the effects of these residues is not trivial, and we do not discuss them here.

In a closely related channel, Mandala and MacKinnon provided cryoEM structures that show actual movement of S4 along the membrane surface. This experiment was performed with a high field of 140 mV across the membrane, produced by a potassium concentration gradient (using liposomes as the membranes, but some liposomes were leaky); unlike the natural membrane, there was no phosphatidyl serine (PS) in the inner leaflet of the membrane, and this was considered to compensate for the high field. They did not show that the putatively closed structure was reversible [112]. However, if PS had been present, there is a strong probability that the charged phosphate group would form a complex with the arginine that had moved out along the membrane surface [63,113]. Such a complex would make it very unlikely for the arginine to return to its original position, ready for another pulse; we therefore consider it probable that this structure is not part of the physiological mechanism by which this channel functions. There is another point concerning these channels that is worth considering. The linker between the voltage-sensing domain and the pore can be cut, and there is still gating [109,114,115]; the explanations that have been proposed do not seem entirely clear. However, if water were to insert between the separated parts, and the linker had been functioning as a proton path, this would be consistent with the proton gating model. These channels are potassium channels, and it has been assumed that they gate in essentially the same manner as domain-swapped channels; the new results suggest that S4-S5 is not a rigid lever. None of the explanations proposed so far include a role for protons, but a water wire could exist even if S4–S5 were severed, although its properties would be altered. Again, the standard model with a rigid, or almost rigid, linker seems not to apply; these channels have up to this point been believed to follow the standard model, but the model must almost certainly be reinterpreted.

### 2.16. Vibrations and Rotations

There is a hydrogen bond network in the K_v_1 family of channels, and it goes through the gate. The network is complex, with complications coming from the fact that there are ions in the gate as well; the ions, being charged, are part of the network, and rearrange it. This statement is based mainly on the following considerations, plus the extensive literature on confined water (Section 2.12 above). Two-dimensional confined water (between graphene sheets) by itself has an altered vibrational spectrum with an additional “fingerprint” peak that has been attributed to what essentially amounts to disruption of the network [116]. A simulation in a confined layer showed how it was possible for such a confined layer of water to act as a pump [117]. With both Cl^−^ and K^+^, the structure is different, but there is little doubt that there are vibrational modes not present in bulk water, or even in ionic solutions. If the gate has a diameter of 1 nm and a thickness of about 1 nm, the volume is approximately 10^−24^ L, which should hold about 30 water molecules at bulk density. This is sufficient for the hydration of perhaps five ions, if every water molecule were in a hydration shell. However, with some water hydrating side chains, probably only two or three ions could be fully hydrated, or perhaps four partially hydrated. There must be ions in the gate region differently hydrated than in the bulk; the structure of the region is, in turn, affected by the ions. Here, the density is different; the ions take up some of the space and interrupt the network. In addition to the ions, the walls of the gate include side chains capable of hydrogen bonding. If the side chains are hydrophilic and polar, they compete with the ions for the water, altering the water dipole orientation. Adding protons to the gate must cause a major rearrangement of the network, including altering the polarity of some side chains. The local hydrogen bond network has to rearrange, probably extensively, to accommodate added protons. This side chain–ion competition plays a major role in dehydrating the ion so that it can pass through the gate; several water molecules must be stripped from the ion hydration shell. Such a rearrangement would necessarily alter the vibrational normal modes of the network. It remains to be shown whether the protonated network has more low-frequency modes; hence, what it does to the partition function for the system, and thus to the free energy, necessarily also remains to be determined. It is not clear how to even begin solving the problem, as the relevant normal modes may involve different amounts of protein. In addition to vibrations, there are rotations, or at least average orientations, that alter the network when protons are added, but rotational degrees of freedom would be more difficult to alter in any way that would have significant thermodynamic effects. The question of whether one has rotations or these are converted to librations remains to be determined, but librations look more probable in the space available. That said, the orientation of the water must change as ions, including protons, move. While much of this paragraph seems redundant, it is worth repeating in the context of vibrations, as these affect thermodynamics, and are, in principle, independently measurable.

### 2.17. Thermodynamic Behavior

The thermodynamic behavior of channels, including K_v_1.2 and Shaker: the previous paragraph suggests the reason for not attempting a direct calculation of the partition function. The possible hysteresis, observed in gating in many cases, would imply the existence of a memory, thus presumably a non-Markovian mechanism [118,119,120,121,122,123,124]. There may be no hysteresis at a steady state [125]. Because the mechanism responsible for this behavior so far is open to multiple interpretations, we do not discuss whether it matters that the gating is out of equilibrium, and even out of a steady state. The fluctuations in current suggest that the gating is a stochastic process; a simple mechanical model would have difficulty in accommodating this. An open state that is a rigid geometrical structure would not allow the observed current fluctuations; any open state would have to be flexible enough to allow the ions to be tied up briefly, followed by allowing them through. Different channels seem to have different fractions of time they shut down in the presumably open state. There have been a number of theoretical investigations that consider what type of model can be associated with the observed fluctuations, but these do not seem to be tied to specific molecular interactions [126,127,128,129,130,131]. A relation to lipid fluctuations has been proposed but has been applied specifically to gramicidin, not a proper membrane channel [132,133]. While there seems to be evidence of an important effect of the lipids on channel behavior for some cases, this has not been tied to a mechanism of voltage gating; there is somewhat better evidence of relevance to mechanosensitive channels, which is to be expected, as membrane distortion would exert a force that could be detected by a mechanosensitive channel. Whether a bundle-crossing model could unbundle if it is held together mechanically is not a question that we have found discussed in the literature so far. Connecting the thermodynamic discussion to a mechanism has not been accomplished yet. Lipid fluctuations may couple to channel fluctuations, and this might apply to both mechanical and proton gating models, but interactions of the lipid are more likely with the VSD.

### 2.18. The Probable Importance of Protons

After this discussion of the probable importance of protons, we have only briefly mentioned, but not really considered, the quantum properties of protons [54,55,56,57,58,59,60]. We did suggest the possibility of tunneling in Section 2.7 above, on the piquito. The proton is a fairly light particle, not as light as an electron, of course, but still small enough to require consideration of its quantum properties. To begin with, the de Broglie wavelength of a proton with thermal energy at about 300 K is approximately 1.6 Å, about a hydrogen bond length. A free proton is known to be able to be transmitted along a water wire, or equivalent; side chains of serine or threonine, ending in -OH, for example, can participate in moving a proton; the same is true of the amide side chains of glutamine and asparagine. Table 1, and its accompanying discussion, shows the conservation of glutamine or asparagine with a serine, generally forming a triad of these amino acids near the gate, or just intracellular to it. A proton can be thought of as delocalized over a finite distance on the scale of bond lengths. Even a water molecule would have a de Broglie length close to 0.4 Å at 300 K.

Gervert and coworkers studied bacteriorhodopsin and found a delocalized proton [134]. Prisk et al. studied rotations of confined water, finding anomalous behavior [135], and an anomalous quantum state of protons in nanoscale confined water was reported by Reiter et al. [72,73]. The delocalized nature of the proton is another reason why classical views of the channel gate and the confined space more generally are uncertain at best. The aqueous crevices in the VSD come within a little more than a 2 Å septum of each other. This may not act as a strong proton barrier, given the properties of the proton; with only a limited proton leak required, a moderate barrier can allow the necessary proton transitions. Within the substantial literature concerning confined water and proton sharing, there are short hydrogen bonds that have a single potential well between oxygens or oxygen and nitrogen. In addition, rings of hydrogen bonds can delocalize a proton, or a pair of protons. We found such a case with a ring with two hydrogen bonds [136]; removing one hydrogen bond destroyed an apparent resonance. Investigations of water and hydrogen bonds, especially in confined spaces, have become an active field of research, and by now show that ignoring quantum effects in such spaces leads to major errors. We cannot review this field in detail, but will just cite several studies that are concerned with the modes of motion of water, proton tunneling, and quantum effects, as well as the relation to the strength and length of hydrogen bonds [97,137,138,139,140,141,142,143,144,145,146,147]. In short, experimental and theoretical evidence points to the importance of quantum effects for hydrogen bonding, water degrees of freedom (rotational and vibrational), and the structure of groups of hydrogen-bonded molecules.

There is another case where the quantum properties of protons, and their delocalization, become important, this time in a non-aqueous solvent, in the calculation of the pK_a_ of acids and bases [148]. While only indirectly relevant in the context of ion channels, it again suggests that ignoring the quantum properties of protons leads to error.

While this section has been devoted to the quantum properties of protons, quantum effects have recently been shown to be important for ion transport in channels [149]. If quantum coherence is important for transport of much larger ions, then it would be surprising if proton properties were not significant in the properties of channels.

### 2.19. Experiments That Appear to Be More Consistent with the Standard Model

All this said, there are experiments that appear to be more consistent with the standard model, at least in the simplest reading of the results. These include a certain number of structures, especially of sodium channels, in which what may be taken as closed (resting) or intermediate structures are either found from cryoEM or are based on models, such as Rosetta models, that are considered to be reasonably reliable. Several of these are based on work with the NaChBac channel, a small bacterial sodium channel, or orthologues of this channel [150,151,152]. These models are interesting but require further examination before they can be considered definitive. For one thing, the interpretations of the models do not consider the possibility that protons might move at all. They test motions of the VSD of NaChBac by coupling cysteines that have been introduced by mutation, and these appear to couple in milliseconds, a very long time in terms of molecular fluctuations. Of course, there is no direct evidence of such fluctuations, so this may be a valid test, but so far, it remains open for further examination. There are structures of some other channels as well, including a potassium channel, of the K_v_4 family [153], showing cryoEM structures that include inactivated and intermediate states, as well as a modeled resting structure. Again, there is no consideration of the possibility of proton motion, and the S4 motions are not large—it is not clear how the putative closed state really couples to closing the gate, although a mechanically closed gate does appear in one cryoEM structure, which may be an inactivated state. We do not rule out the possibility that some protein rearrangements may occur; certainly side chain motions exist, and we find them in our own calculations. If a side chain rotates into a passage, thus covering 2 Å, and this happens from at least two of the four domains, a passage that had been 5 Å could now become 1 Å, hence mechanically closed without S4 motion in a cryoEM structure. Such cases do not appear to have been reported. If the rotation is effected by the motion of a proton, this remains conceivable. However, this does not appear to be consistent with the dehydration mechanism for which the evidence is stronger. Nevertheless, it does show how a proton-controlled gate could be responsible for a geometric, mechanical gate closing—in principle. We do not see evidence that this actually happens.

### 2.20. Energetics

This is an important topic that we have left for last, because it is subject to many alternate interpretations. Section 2.17 (thermodynamics) could not come to firm conclusions. In the end, we will be able to understand the energetics in detail only after we have a more complete mechanism for gating. We begin to understand the complexity by looking back at Figure 4, showing the K_v_2.1 gate. If the ion rehydrates as it moves past the constriction at the prolines, into the cavity, there is no net cost of dehydration at the gate, only an activation energy; the net energetic cost has to be paid at the selectivity filter, where the protein solvates the ion. Since the ion is going from high concentration intracellularly to low concentration outside, there is a net driving force for the ionic current, in addition to the electric field, coming from the dilution of the ions. A relatively early study suggested the complexity of the problem [154]. An important new study considers the effect of a temperature step at several different initial temperatures, with mutants that alter or abolish the gating, apparently through the linker [155]. The authors attach their findings to several possible transitions. However, they do not consider the possible effects on hydrogen bonds, which collectively could account for about the same magnitude of energy transitions. At this point, we do not have enough information about the energetics to allow inferences with regard to mechanism, because more than one mechanism could account for the temperature behavior. The values of Q10 reported earlier [154] suggest that several processes occur sequentially, some (especially a relatively early transition) with large Q10 > 4, some with ionic current Q10 = 1.2, and it is not possible to attribute the steps to specific transitions. What can be stated at this time is that there are several steps in gating involving different parts of the channel. There are different steps that are compatible with more than one step in the gating mechanism, and the order of magnitude of the temperature effects that are observed can be consistent with more than one mechanism.

### 2.21. Omissions from This Review

We have not discussed the final two topics that are possibly relevant. First, lipids have been reported to affect gating, and second, we have very little discussion of the current–voltage curve, which looks like that of a standard rectifier; it shows how an external potential, in the conducting direction, produces a current roughly proportional to the potential. The reason for omitting the lipids is that it appears, from our reading of the literature, that the lipids might influence the probability of producing protons or a proton cascade in the VSD, rather than alter the network of interactions at the gate. The lipids do not appear to stretch all the way to the gate entrance; however, S4 of the VSD may interact with the lipid; the other transmembrane segments may also interact. These interactions are not relevant to the question we are dealing with here, at least not directly relevant. Second, the potential may increase current by affecting the probability of transitions among local minima required for an ion to move forward in the alreadyss open configuration of the gate. Here, we are concerned with the closed → open transition. In the section on dehydration, we did briefly consider the effect of the electric field on dehydration [107].

## 3. Summary

There are a number of experiments that are consistent with proton gating; some of these have been interpreted as consistent with the standard model of gating, in which S4 moves to close the channel pore through a mechanical linker; others have been ignored, perhaps to be interpreted at some time in the future. As far as we can see, all these experiments can be interpreted in terms of the proton gating model. There is an experiment that is consistent with proton tunneling to start a possible proton cascade; the H_v_1 channel shows that proton transport is possible in the VSD, and a somewhat similar path exists in the M2 channel. Experiments seemed to show cysteine access to the membrane surface, so that they could relate to MTS, but they do not prove that S4 has physically moved. Molecular dynamics simulations have serious defects, not least that they often use an inadequate model for water and do not allow charge transfer or bond breaking. Such MD simulations that attempt to simulate gating require unrealistic potentials; with realistic potentials, S4 does not move. It is also hard to see how these could be reversible upon removal of the voltage. Quantum effects cannot be taken into account in a classical simulation, but the dimensions of the gate and the size of the de Broglie wavelength of a thermal proton suggest that ignoring these effects will lead to serious error. Finally, the fact that forms of closed state can exist without S4 motion, for example in the EAG channel, and in KirBac 1.1, means that S4 motion is not required for gating; deciding which gating mechanism is correct in the physiological case is a matter for further experiment. The proton gating model is consistent with some specific experiments that have not been explained with the standard models: substitution of serine and of aspartate at the gate, the effect of D_2_O, the “piquito”, and the effect of T1. The standard model, with S4 motion, in most of its forms, does not consider the water at the membrane surface, or in the gate region, in appropriate detail. At this point, the proton gating model is therefore at least plausible, and seemingly more likely to be correct than the standard model; the role of protons and hydration must be taken into account in the interpretation of all data regarding ion channel structure and function.

## Figures and Tables

**Figure 1 ijms-26-07325-f001:**
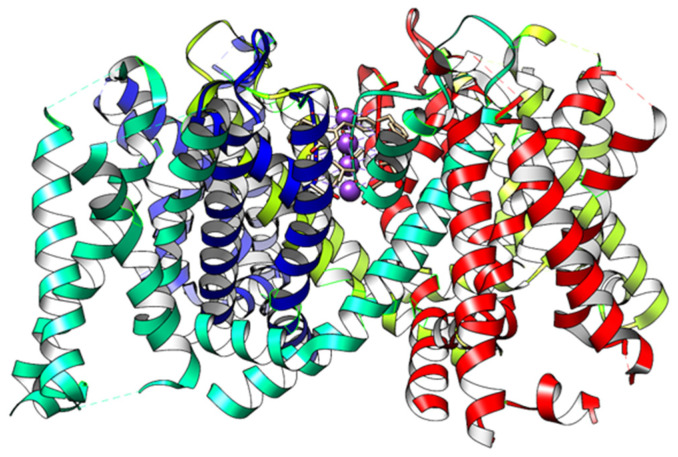
A traditional view of the channel structure shows only protein, but ignores the water and ions that are required for it to function. This version, created by Chimera from the K_v_1.2 (pdb: 3Lut) open structure, shows the 4 voltage-sensing domains and includes K^+^ ions in the selectivity filter—ions in all 4 positions, although they are almost certainly not all occupied simultaneously. The X-ray structure does include water molecules, but they are not included by Chimera.

**Figure 2 ijms-26-07325-f002:**
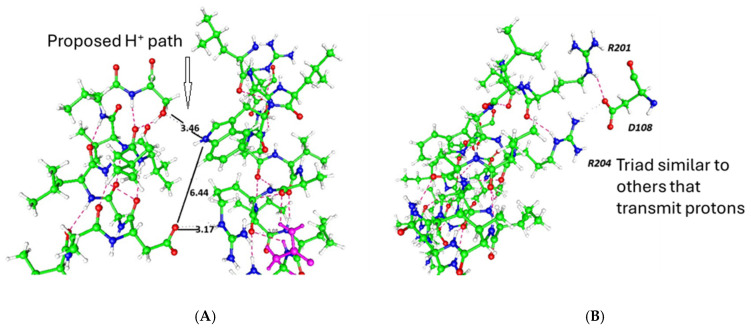
A view of part of the H_v_1 channel. Carbon atoms are green, oxygen red, nitrogen blue, and hydrogen white. (**A**) The proposed proton path (black line) is shown with distances in Angstroms as the polar residues connect. The 3 Å distances are appropriate for long hydrogen bonds, while 6 Å is essentially exactly what is needed for one water molecule. (**B**) Another view with the figure rotated and reaching slightly further to the extracellular side, showing an arg–asp–arg triad that appears to be part of the proton path (coordinates from pdb 3WKV). Dashed lines indicate hydrogen bonds, as inserted by the gOpenMol-3.00 program.

**Figure 3 ijms-26-07325-f003:**
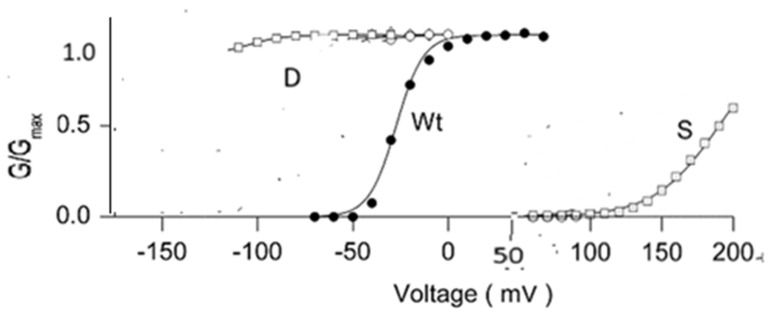
Adapted from Figure 3B and 3C in the study by Sukhareva et al. [33]; the current voltage curves of wild type (WT) and the P475X mutations (X=D or S) of the Shaker channel, showing that the aspartate mutation is open at all physiologically relevant potentials, essentially constitutively open, while the serine mutation is closed at all physiologically relevant potentials. Several other mutations fell between these limits, much closer to WT. The change in scale of the horizontal axis at +50 mV has no significance, The aspartate line (D) extends almost completely horizontally to −150 mV in the original. (we thank Dr. K. Swartz for permission to copy his data; presentation in this form, however, is the responsibility of the present authors). Not only must the ion be dehydrated at the gate, but it is then rehydrated in the cavity just beyond the gate [36]. This makes sense, as the cavity contains multiple water molecules. These are available for rehydrating the ion as it emerges from the gate region; the hydration is not so strong that the ion cannot again be partially dehydrated as it enters the selectivity filter at the opposite end of the pore from the gate. The driving force for the ion as it passes through the pore must include an accounting for the role of water.

**Figure 5 ijms-26-07325-f005:**
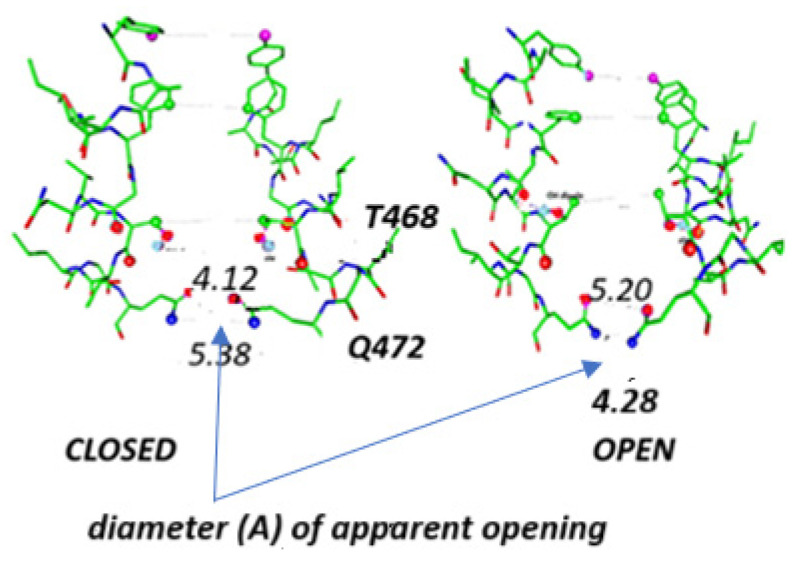
The closed (pdb:7YID) and open (pdb: 7YIF) gates of the EAGchannels, as determined by Zhang et al. Note that the effective opening is essentially the same in both, with only the shorter interdomain distance due to the orientation of the glutamine, Q472. However, in both cases, the distance is about 4+ Å. The gating does not appear to be a mechanical closure of the gate. There is also reorientation of the threonine hydroxyl above the glutamine, but this also does not constrict the path of the ion.

**Table 1 ijms-26-07325-t001:** Examples of proton transmitting amino acid triads *.

Channel	Amino Acids—Residue Number	pdb Code
K_v_1.2	N-412; S-411; N-414	3Lut
K_v_2.1	S413; N410; S413 (next domain)	8SD3
Shaker	N480; S479; N482	7SIP
H_v_1	Q219; S215; Q98	3WKV
	Y157; D170; R207	
Bacteriorhodopsin	Y57; R82; Q194	1FBB
	Y185; W86; R212	

* H_v_1 and bacteriorhodopsin have more than one relevant triad; also, one can find a square arrangement of these amino acids in some channels, but it is not clear that these are involved with proton transfer. The gate must be regarded as a three-dimensional region, rather than a planar set of prolines. The triads are found in H_v_1 channels, where they are explicitly seen as part of a proton path, and other channels as listed in Table 1. There is no obvious reason that proton-transmitting moieties should be present in so many channels if there is no need to transmit protons.

## Data Availability

There is no original data in this review; the references are all listed.

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
