# Peer review of "H+ and Confined Water in Gating in Many Voltage-Gated Potassium Channels: Ion/Water/Counterion/Protein Networks and Protons Added to Gate the Channel"

_ijms, 2025, doi:10.3390/ijms26157325_

Round 1
Reviewer 1 Report
Comments and Suggestions for Authors
The review “H+ and confined water in gating in many voltage gated potassium channels: ion /water/counterion/protein networks, and added protons to gate the channel” is an interesting account of data and thoughts rarely discussed in mainstream papers on voltage gating.
The authors extensively discuss the roles of protonation and proton movement in gating, implicate water dynamics and solvation processes in the confinement of the channel’s cavities, and challenge the absolute necessity of significant movement of the voltage sensor S4 helix for the opening mechanism. They also present the vulnerabilities of cysteine accessibility analysis and the limitations of current MD. The authors mention that experimentally observed single-channel fluctuations lead to kinetic schemes where the transitions are rarely connected to specific molecular interactions. These are healthy analyses and discussions of existing data that, in my view, need to be seen by the community. The text is mainly easy to read, but has repetitions, so it can potentially be tightened.
I have a few suggestions on how to improve the readability and better illustrate the narration.
The authors compare the Hv1 proton channel with the common V-gated (K+, Na+) channels in several places. In the end, they also propose an evolutionary path for creating a voltage-gated channel by fusing Hv1 (acting as VSD) with KcsA (providing an ion conductive path).
I suggest that the authors show the structure of Hv1, indicating a hypothetical (or specific) path for proton movement. As discussed by DeCoursey (J. R. Soc. Interface, 2018), it is still difficult to untangle proton motion from the Hv1 gating process, yet this is the best example where proton motion exerts an effect on the functional state of a channel. It may, indeed, involve minimal displacements of S4. In the next panel, I would suggest presenting the structure of the most appropriate Kv channel (Table 1), with locations of titratable sidechains near the gate forming proton-conducting triads or larger networks.
More particular points:
L143-146. It would be desirable to illustrate the proton cascade in a figure.
L197-200. It is difficult to exclude backbone motion, but sidechain reorientations can certainly be involved. The question is in the huge time disparity between these two types of rearrangements. Each particular case should be related to the experimental kinetics of the processes in question.
L209-214. While explaining the problem of ion desolvation, it may be helpful to discuss existing data on the residence time of water in the first hydration shell around the ion and the impact of competition with protein sidechains.
L257-269. The estimations supporting the proton tunneling hypothesis are unclear. This part needs to be better explained.
L353-362. The authors speak about the lack of reversibility in MD simulations. They should take into account VERY short simulation times compared to real processes and shallow energy gradients. I would not discard simulations that reasonably predict only the forward path.
L423-427. The message of this short section needs to be better explained.
Fig. 3. The context of these groups of residues remains unclear, and there are no indications on the figure of what corresponds to what. The proton-transmitting triads should be depicted more clearly, and the spatial relationship with the cytoplasmic side of the channel also needs to be illustrated.
Figures 4 and 5 also need similar improvement.
Author Response
The review “H+ and confined water in gating in many voltage gated potassium channels: ion /water/counterion/protein networks, and added protons to gate the channel” is an interesting account of data and thoughts rarely discussed in mainstream papers on voltage gating.
The authors extensively discuss the roles of protonation and proton movement in gating, implicate water dynamics and solvation processes in the confinement of the channel’s cavities, and challenge the absolute necessity of significant movement of the voltage sensor S4 helix for the opening mechanism. They also present the vulnerabilities of cysteine accessibility analysis and the limitations of current MD. The authors mention that experimentally observed single-channel fluctuations lead to kinetic schemes where the transitions are rarely connected to specific molecular interactions. These are healthy analyses and discussions of existing data that, in my view, need to be seen by the community. The text is mainly easy to read, but has repetitions, so it can potentially be tightened.
We agree that the prose needed improvement, and there has been extensive rewriting.
I have a few suggestions on how to improve the readability and better illustrate the narration.
The authors compare the Hv1 proton channel with the common V-gated (K+, Na+) channels in several places. In the end, they also propose an evolutionary path for creating a voltage-gated channel by fusing Hv1 (acting as VSD) with KcsA (providing an ion conductive path).
I suggest that the authors show the structure of Hv1, indicating a hypothetical (or specific) path for proton movement. As discussed by DeCoursey (J. R. Soc. Interface, 2018), it is still difficult to untangle proton motion from the Hv1 gating process, yet this is the best example where proton motion exerts an effect on the functional state of a channel. It may, indeed, involve minimal displacements of S4. In the next panel, I would suggest presenting the structure of the most appropriate Kv channel (Table 1), with locations of titratable sidechains near the gate forming proton-conducting triads or larger networks.
There is now an Hv1 figure with a proton path proposed as noted on the figure. This covers only part of the channel but shows how the proton has a path through a significant section of the channel. Table 1 has proton transmitting triads from several channels. Also, we published earlier papers (refs 3 – 6). In essence what the reviewer suggests has been partially already published. However, rather than an extensive repeat of our previously published work, which might extend the size of the paper tremendously, we refer to the published work. The new Hv1 figure (in two parts) also helps respond to this point.
More particular points:
L143-146. It would be desirable to illustrate the proton cascade in a figure. This refers to the VSD, and a reference is given, rather than repeat the previous paper.
L197-200. It is difficult to exclude backbone motion, but sidechain reorientations can certainly be involved. The question is in the huge time disparity between these two types of rearrangements. Each particular case should be related to the experimental kinetics of the processes in question. We agree that side chain motions can be important, and in fact discuss this. The only way in which the time scale is really important in this question is whether these motions are faster than the gating process or not. Side chain motions are faster, and that is all that really is important here. The exception is the piquito, where we do discuss time scales.
L209-214. While explaining the problem of ion desolvation, it may be helpful to discuss existing data on the residence time of water in the first hydration shell around the ion and the impact of competition with protein sidechains. Here the reviewer has touched on an important point that is hard to answer. The time scale for hydration and dehydration is in the tens of picosecond range in bulk solution. It is much slower in confined water. We do discuss as much as we can glean from the literature on this point, but we agree that it is important. A couple of new references are cited that are concerned with this question. As to competition with the protein side chains, this is of unquestioned importance. However, the detailed mechanism of dehydration is somewhat less important than the fact that it must occur (the ion enters the gate with 6 waters of hydration and leaves with no more than 2, as shown, for example, by streaming current measurements. The hydration of side chains, the effect of counterions, and the effect of additional protons in the closed state, are matters on which we are now carrying out computations. We hope to have these completed before too long, but they are not yet ready for publication. In this review article, we bring the story up to date as it stands with already published literature.
L257-269. The estimations supporting the proton tunneling hypothesis are unclear. This part needs to be better explained. We have done as much as we can with this, short of postulating a barrier that would fit the data, but for which there is no direct evidence. At this point, we can say that a barrier of the required form can exist; if no barrier could plausibly be postulated, then the tunneling could be ruled out. For example, tunneling by K+ appears impossible under any plausible scenario. For protons, we note that we could create such a barrier under conditions that could exist in the channel. This is important, but we cannot go farther without more direct evidence.
L353-362. The authors speak about the lack of reversibility in MD simulations. They should take into account VERY short simulation times compared to real processes and shallow energy gradients. I would not discard simulations that reasonably predict only the forward path. Simulation times now are longer than gating times. Starting from the open state and going to the closed state should be possible, as well as the reverse. Closed to open requires an added potential, but what is generally used is >>70 mV, and no simulations, to the best of our knowledge, have accomplished reversing this. We assume the reviewer means open to closed as forward, as the simulations are done with voltage on, and coordinates of the open state are known.
L423-427. The message of this short section needs to be better explained. We are not sure what this refers to, but there has been appreciable rewriting of the entire paper in the interests of clarity.
Fig. 3. The context of these groups of residues remains unclear, and there are no indications on the figure of what corresponds to what. The proton-transmitting triads should be depicted more clearly, and the spatial relationship with the cytoplasmic side of the channel also needs to be illustrated.
Figures 4 and 5 also need similar improvement.
On the figures, the reviewer is correct. Fig 3 has been replaced, and two new figures added. Labeling showing proton paths has been included, and new captions added. On rereading the paper, we also came to the conclusion that these figures needed improvement.
Reviewer 2 Report
Comments and Suggestions for Authors
This review describes the authors’ hypothesis that the mechanism by which voltage-gated channels open and close involves proton movement, rather then movement of the channel protein.
Specific points:
A general question is whether the authors agree (and I suspect that they do) that it is likely and perhaps inevitable, that membrane potential changes will result in the movement of charges intrinsic to the protein. Perhaps this point could be made in the paragraph starting line 69. It could be said that certainly the protein moves, but this movement may not be responsible for gating.
In lines 132-134 (Section 1.1), it is suggested that the existence of a dimeric form of Hv1 somehow is a significant difference from the VSD of K channels. It should be noted that Hv1 can exist as a monomer which behaves very similarly to the dimer, with some subtle differences in gating kinetics.
Section 1.7 about the piquito, feels a bit fuzzy. Is there some reason this might not simply be a small movement of part of the protein, rather than a proton? Even if it does reflect proton tunneling, if it carries only 1% of the gating current, it does not appear to provide very strong support for the idea that a large part of the gating charge is attributable to proton movement. I feel I am missing some key part of the argument. Is the idea that the piquito is too fast to reflect movement of a charge within the protein? Is the idea that if this part of the gating charge can be ascribed to proton movement, this supports the idea that the rest could also?
Section 1.8 discusses the MTS/Cys interaction. To my great surprise, in the first Yang & Horn (1995) paper the authors do in fact discuss Cys being at the membrane surface. However, by the third paper (Yang et al, 1997; Probing the outer vestibule of a sodium channel voltage sensor.
Yang N, George AL Jr, Horn R. Biophys J. 1997 Nov;73(5):2260-8. doi: 10.1016/S0006-3495(97)78258-4), they are clearly envisioning (see their Figs. 1 & 6) a channel with aqueous vestibules that allow MTS reagents to enter the pore and travel partway through the channel, reaching what is now called a hydrophobic gasket or charge transfer center that separates those parts of the pore that are internally accessible from those that are externally accessible. This idea was very important and transformational in the history of voltage-gating, because it revealed the fact that most of the voltage drop across the channel ocurrs over a short hydrophobic region (as opposed to dropping linearly with distance through the membrane). Later studies by Starace & Bezanilla, using Arg->His mutations in S4, confirmed that the K channel VSD has aqueous access much of the way across, with a very short region that excludes water (e.g., Fig. 5 in Starace DM, Bezanilla F. A proton pore in a potassium channel voltage sensor reveals a focused electric field. Nature. 2004 Feb 5;427(6974):548-53. doi: 10.1038/nature02270).
The critique of MD (Section 1.10) is insightful. Certainly MD does not include protonation/deprotonation events and thus, by definition, will be oblivious to such phenomena.
Section 1.14 (eag) presents a strong argument.
Section 1.15 begins with an ambiguous sentence. Which channel is meant by “the channel” and where is “the gate”? The preceding section discussed EAG channels, so can we assume this is “the channel” or have we moved on to another channel? The only two references in this section (102, 103) refer to channels generically – is this all meant to be true for all channels? But how can that be possible, since the gate differs for different channels?
Section 1.16 “Thermodynamics” compounds the same question. Which channels comprise “these channels” [line 572]? The references cited here cover a wide array of channels. Are we now discussing all channels, or some unspecified subset (e.g., voltage-gated channels)?
Line 580 mentions “oscillations in current” suggesting stochastic gating. Should this rather be “fluctuations” or “noise” in current? Oscillations indicate a regular pattern, which contradicts stochastic behavior. Also, does this refer to changes in the current through a single open channel, or to the fluctuations of total current due to the stochastic opening and closing of a population of channels?
Author Response
Reviewer 2
Comments and Suggestions for Authors
This review describes the authors’ hypothesis that the mechanism by which voltage-gated channels open and close involves proton movement, rather then movement of the channel protein.
Specific points:
A general question is whether the authors agree (and I suspect that they do) that it is likely and perhaps inevitable, that membrane potential changes will result in the movement of charges intrinsic to the protein. Perhaps this point could be made in the paragraph starting line 69. It could be said that certainly the protein moves, but this movement may not be responsible for gating. The gating current is a measured quantity—positive charge moves, but it does not have to be the arginines from S4 of the VSD. Side chains may rotate or otherwise reorient, so if this is what the reviewer means, we agree. However, protons can also move, and account for the measured gating current
In lines 132-134 (Section 1.1), it is suggested that the existence of a dimeric form of Hv1 somehow is a significant difference from the VSD of K channels. It should be noted that Hv1 can exist as a monomer which behaves very similarly to the dimer, with some subtle differences in gating kinetics. We have done this.
Section 1.7 about the piquito, feels a bit fuzzy. Is there some reason this might not simply be a small movement of part of the protein, rather than a proton? Even if it does reflect proton tunneling, if it carries only 1% of the gating current, it does not appear to provide very strong support for the idea that a large part of the gating charge is attributable to proton movement. I feel I am missing some key part of the argument. Is the idea that the piquito is too fast to reflect movement of a charge within the protein? Is the idea that if this part of the gating charge can be ascribed to proton movement, this supports the idea that the rest could also? This is not certain proof, but if it is a random wiggle, there is no reason why it shouldn’t happen randomly. The reason we consider it significant is that it could start a cascade of protons if the proton moves a short distance, and the system is set up to have this contribution to the current, albeit small, trigger the remaining part of the gating current. In the standard models, there is no apparent reason at all for the piquito to exist; 1% of the gating current would correspond to about a 3 A movement for a single charge, which makes sense for a tunneling proton, but is a little distant for an arginine side chain (the only chain with a charge—but the most important point is that such a movement, in the context of an S4 motion, is pointless. In a standard model the piquito should not exist—but it does). We have not found a change in bond angle that seems consistent with such a motion, nor any obvious means to couple such a twist to the remainder of the gating current. Sigg and Bezanilla made a qualitative attempt to account for the piquito with a hypothetical potential landscape; however, no such landscape has been shown to be plausible for the structure of a VSD. [D. Sigg, H. Qian and F. Bezanilla Biophys. J. 1999 Vol. 76 Issue 2 Pages 782-803]
Section 1.8 discusses the MTS/Cys interaction. To my great surprise, in the first Yang & Horn (1995) paper the authors do in fact discuss Cys being at the membrane surface. However, by the third paper (Yang et al, 1997; Probing the outer vestibule of a sodium channel voltage sensor. (Yang N, George AL Jr, Horn R. Biophys J. 1997 Nov;73(5):2260-8. doi: 10.1016/S0006-3495(97)78258-4), they are clearly envisioning (see their Figs. 1 & 6) a channel with aqueous vestibules that allow MTS reagents to enter the pore and travel partway through the channel, reaching what is now called a hydrophobic gasket or charge transfer center that separates those parts of the pore that are internally accessible from those that are externally accessible. This idea was very important and transformational in the history of voltage-gating, because it revealed the fact that most of the voltage drop across the channel ocurrs over a short hydrophobic region (as opposed to dropping linearly with distance through the membrane). Later studies by Starace & Bezanilla, using Arg->His mutations in S4, confirmed that the K channel VSD has aqueous access much of the way across, with a very short region that excludes water (e.g., Fig. 5 in Starace DM, Bezanilla F. A proton pore in a potassium channel voltage sensor reveals a focused electric field. Nature. 2004 Feb 5;427(6974):548-53. doi: 10.1038/nature02270). 1) This paper is interesting in that it shows that protons can traverse the voltage sensing domain. The focused electric field is foreshadowed by a paper the previous year from the Bezanilla group (Asamoah et al reference 46 ) that showed the entire 70 mV transmembrane voltage dropping across a single arginine--a field of at least 108 V m-1 dropping across 7 A would be 70 mV. There should be no voltage drop across the rest of the VSD. 2) The comment however focuses on accessibility of the cysteines that have replaced the arginines. These presumably have access through the crevices, which are considerably enlarged by the arg àcys mutation, considering the much smaller size of the cys. Considering that the size of the barrier between the crevices is small to begin with, it is too difficult to interpret the cys results at all, which is what we intended to say.
The critique of MD (Section 1.10) is insightful. Certainly MD does not include protonation/deprotonation events and thus, by definition, will be oblivious to such phenomena. We obviously agree.
Section 1.14 (eag) presents a strong argument. We obviously agree.
Section 1.15 begins with an ambiguous sentence. Which channel is meant by “the channel” and where is “the gate”? The preceding section discussed EAG channels, so can we assume this is “the channel” or have we moved on to another channel? The only two references in this section (102, 103) refer to channels generically – is this all meant to be true for all channels? But how can that be possible, since the gate differs for different channels? This is my (MEG) fault. The writing was ambiguous. This has been rewritten
Section 1.16 “Thermodynamics” compounds the same question. Which channels comprise “these channels” [line 572]? The references cited here cover a wide array of channels. Are we now discussing all channels, or some unspecified subset (e.g., voltage-gated channels)? This is my (MEG) fault. The writing was ambiguous. This has been rewritten
Line 580 mentions “oscillations in current” suggesting stochastic gating. Should this rather be “fluctuations” or “noise” in current? Oscillations indicate a regular pattern, which contradicts stochastic behavior. Also, does this refer to changes in the current through a single open channel, or to the fluctuations of total current due to the stochastic opening and closing of a population of channels? Again, this has been rewritten; it now reads fluctuations.
Reviewer 3 Report
Comments and Suggestions for Authors
The manuscript proposes an alternative hypothesis to the widely accepted paradigm of voltage-gated potassium (Kv) channel gating, challenging the central role of conformational movement of the S4 segment. Instead, it suggests that the channel opening and closing process could be primarily mediated by: mobile protons (H⁺), confined water, and cooperative networks of ionic interactions, protons, and protein side chains.
This alternative model emphasises the role of quantum phenomena such as proton tunnelling and partial dehydration of K⁺ as fundamental processes in gating.
Novelty assessment:
-
(i) It is original in its attempt to reinterpret classical evidence from a protonic and cooperative perspective.
-
(ii) It is not entirely new: the authors have previously published similar hypotheses. However, the manuscript serves as a critical argumentative synthesis that could be valuable to the scientific community.
Key issues identified:
-
Absence of a methodological section, despite references to simulations and quantitative arguments.
-
Lack of clear diagrams or figures illustrating the alternative model.
-
No comparative table between the conventional and proposed models, which hinders comprehension for general readers.
Recommended improvements:
-
Add a methodology section: Even if theoretical, it should specify which simulations were performed, the level of theory (if applicable), and the rationale behind the model.
-
Clarify the argumentative structure: Reorganise the manuscript into sections such as classical hypothesis, contradictory evidence, alternative model, theoretical validation, and future projections.
-
Include conceptual figures: For example, comparisons between the S4 displacement model and the H⁺/H₂O/side chain network model.
-
Remove unfounded evolutionary speculations (e.g., section 1.20).
-
The proton-based model is overemphasised without structural validation: the S4 movement is dismissed as artefactual without direct experimental evidence to refute it.
-
While the criticisms of molecular dynamics (MD) simulations are valid, the manuscript fails to mention recent advances such as polarisable water models or ab initio approaches (e.g., polarisable continuum models, Drude oscillators).
-
The hypothesis of proton tunnelling is not supported by energetic calculations or transition state models.
-
The energetic cost of K⁺ dehydration — a key issue in selective transport — is omitted from the analysis.
The overall quality of English in the manuscript is acceptable but requires significant improvement to meet the standards of an international scientific journal. The text contains numerous grammatical errors, awkward sentence constructions, and inconsistent use of technical terminology. Common issues include incorrect verb tenses, improper punctuation, and unnatural phrasing that can obscure the intended meaning. Additionally, the overuse of conditional language and vague expressions weakens the clarity and strength of the arguments. A thorough revision by a native English speaker or a professional scientific editor is strongly recommended to enhance readability and ensure precise scientific communication.
Author Response
Reviewer 3
Comments and Suggestions for Authors
lvavoltage-gated potassium (Kv) channel gating, challenging the central role of conformational movement of the S4 segment. Instead, it suggests that the channel opening and closing process could be primarily mediated by: mobile protons (H⁺), confined water, and cooperative networks of ionic interactions, protons, and protein side chains.
This alternative model emphasises the role of quantum phenomena such as proton tunnelling and partial dehydration of K⁺ as fundamental processes in gating.
Novelty assessment:
- (i) It is original in its attempt to reinterpret classical evidence from a protonic and cooperative perspective.
- (ii) It is not entirely new: the authors have previously published similar hypotheses. However, the manuscript serves as a critical argumentative synthesis that could be valuable to the scientific community.
Key issues identified:
- Absence of a methodological section, despite references to simulations and quantitative arguments. This is a review article. We cannot give the methods of over 150 papers—review articles generally do not have methods sections.
- Lack of clear diagrams or figures illustrating the alternative model. Two new figures have been added; the relevant figures are now labeled, with distances and connections labeled. One figure which added little has been removed. The reviewer is essentially correct on this point. We also note that our previous publications, cited (refs 3-6) cover some of this matter.
- No comparative table between the conventional and proposed models, which hinders comprehension for general readers. While the request for clear figures makes perfectly good sense, it is not clear what a Table would consist of. We have not added a Table. The difference between models consists of a major qualitative difference between gating current consisting of large S4 movement and movement of protons. Details may vary, but there is no question as what the difference between models consists of.
Recommended improvements:
- Add a methodology section: Even if theoretical, it should specify which simulations were performed, the level of theory (if applicable), and the rationale behind the model. There are multiple simulation papers referred to, each with a force field, and a version of a force field and the reason for choice of water model, choice of amount of water and lipid to include, and many other details—all of which constitute the rationale for the particular paper. It is impossible to repeat all the methodological sections of all the papers cited. We have discussed what these papers have in common that makes their results questionable. If the simulation methods are all described, the experimental references would have to be included in the methods section. We are only writing a review article, not a complete book.
- Clarify the argumentative structure: Reorganise the manuscript into sections such as classical hypothesis, contradictory evidence, alternative model, theoretical validation, and future projections. I am not sure what kind of paper this would be—I don’t think it makes any sense in this case. For example, one section of the article now discusses the effects of substituting D2O. There is no contrary evidence. The evidence is what it is. The D2O set of experiments gets its own section. Then it is necessary to move on. In those sections that have possible alternate explanations, such as the “Piquito”, that point is made. The organization of the paper is by category of evidence, with alternate interpretations given in each section. There is a section devoted to the primary evidence that gave rise to the standard model; this section too has its own set of alternate explanations, in this case explanations that are consistent with the proton gating model, together with the original interpretations coming from the standard model. There are experiments, and attempts to explain them. In a sense, the present organization comes as close as is possible to what the reviewer suggests. Future projections are beyond the scope of this article as many groups each add to our knowledge. We cannot guess what every worker in the field will think of.
- Include conceptual figures: For example, comparisons between the S4 displacement model and the H⁺/H₂O/side chain network model. There are several versions of the standard (S4) model, expounded in numerous papers, some of which are cited here. Working through the details of at least most of the proposals in the literature, and commenting on each, is not possible. Here we agree with the reviewer that the figures in our original submission were inadequate. The revised version has new figures, with more detailed labels. However, the new version remedies this.
- Remove unfounded evolutionary speculations (e.g., section 1.20). This section is interesting, but not really part of the central argument, and is therefore withdrawn
- The proton-based model is overemphasised without structural validation: the S4 movement is dismissed as artefactual without direct experimental evidence to refute it. This comment, in effect, says the paper should not be written, and the new model should not be considered. Evidence against S4 motion is given section by section—it is what the entire paper consists of. We believe it is time to summarize the evidence that supports the proton gating model; this model is not “overemphasized”; it is the subject of the paper. The evidence, section by section, is given in as much detail as seems appropriate. The paper acknowledges that the proton model contradicts what is generally believed, and gives the reasons for reconsidering this general belief.
- While the criticisms of molecular dynamics (MD) simulations are valid, the manuscript fails to mention recent advances such as polarisable water models or ab initio approaches (e.g., polarisable continuum models, Drude oscillators). In the new version of the paper, this comment is taken into account, with suitable citations; however, the primary criticisms remain, with reasons given.
- The hypothesis of proton tunnelling is not supported by energetic calculations or transition state models. This comment, unfortunately, is true. Energetic calculations are best derived from data on temperature dependence. There is a fair amount of data in the literature giving the Q10 values in the literature. The one thing that is clear Is that the there are several steps, and the overall process is complex. This point is made in the paper, and discussed to the extent present data allows. We agree with the reviewer that understanding the energies involved would be extremely valuable. There is a section in the paper on energetics, but it really comes to no conclusions—it does acknowledge the importance of the topic. We are now doing calculations that will, among other things, partially address this matter, but these calculations are not yet ready for publication.
- The energetic cost of K⁺ dehydration — a key issue in selective transport — is omitted from the analysis. Here the reviewer has a point. We have added some references; the problem is that dehydration energy depends on the environment; the cost of dehydration is well known in vacuum; it is known on average in bulk water. However, it is not so clear in confined water, although we discuss the question to the extent possible. The added references help with this discussion. It is, as the reviewer notes, an important question. However, the fact of dehydration is not in question, even in the standard model; the K+ enters the gate of the channel with six hydrating water molecules, and leaves with at most two, so four have been removed. How this happens, however, is an important question, and we are not entirely ready to answer it. We are doing calculations now ourselves to solve this question, and hope to publish the results later this year. However, for this review article, there is not enough data for a discussion. We do agree with the reviewer that this is an important question.
Comments on the Quality of English Language
The overall quality of English in the manuscript is acceptable but requires significant improvement to meet the standards of an international scientific journal. The text contains numerous grammatical errors, awkward sentence constructions, and inconsistent use of technical terminology. Common issues include incorrect verb tenses, improper punctuation, and unnatural phrasing that can obscure the intended meaning. Additionally, the overuse of conditional language and vague expressions weakens the clarity and strength of the arguments. A thorough revision by a native English speaker or a professional scientific editor is strongly recommended to enhance readability and ensure precise scientific communication.
My (MEG wrote the paper, and this comment is written by him) first reaction to this comment was very negative. I am a native speaker, and I use correct English grammar. However, on rereading the paper, I can understand what the reviewer is saying. There were typos, a number of which were omitted words; the omissions created sentences with grammatical errors. I can see why the reviewer took the English to be inadequate. In addition, I use a number of semicolons, and in small type, these may be mistaken for commas, leading to sentences that appear to be comma-spliced, another error. Hopefully, the paper will appear with the semicolons clear. I am not going to remove them, as they help the prose flow by connecting related thoughts. This said, a major rewrite was called for. The original version was redundant and difficult to read or understand. I have rewritten much of the manuscript, with changes in almost every paragraph. There is some remaining redundancy, as a few questions have to be considered from more than one point of view, so those points are repeated. In addition, in the original version, I left some pronouns with ambiguous antecedents. These have all been disambiguated in the present version. I thank the reviewer for insisting on a rewrite, although I would have picked this up myself on rereading the manuscript. I am not quite sure how I let the original go in poor form, but I do not need an editor to tell me that the first version needed to be rewritten.
v
Round 2
Reviewer 3 Report
Comments and Suggestions for Authors
All figures are poor quality. We request that the images be sharpened. In some cases, the molecules are not clearly distinguishable.
Comments on the Quality of English LanguageEnglish is good, but can improve.
Author Response
Comments and Suggestions for Authors
lvavoltage-gated potassium (Kv) channel gating, challenging the central role of conformational movement of the S4 segment. Instead, it suggests that the channel opening and closing process could be primarily mediated by: mobile protons (H⁺), confined water, and cooperative networks of ionic interactions, protons, and protein side chains.
This alternative model emphasises the role of quantum phenomena such as proton tunnelling and partial dehydration of K⁺ as fundamental processes in gating.
Novelty assessment:
- (i) It is original in its attempt to reinterpret classical evidence from a protonic and cooperative perspective.
- (ii) It is not entirely new: the authors have previously published similar hypotheses. However, the manuscript serves as a critical argumentative synthesis that could be valuable to the scientific community.
Key issues identified:
- Absence of a methodological section, despite references to simulations and quantitative arguments. This is a review article. We cannot give the methods of over 150 papers—review articles generally do not have methods sections.
- Lack of clear diagrams or figures illustrating the alternative model. Two new figures have been added; the relevant figures are now labeled, with distances and connections labeled. One figure which added little has been removed. The reviewer is essentially correct on this point. We also note that our previous publications, cited (refs 3-6) cover some of this matter.
- No comparative table between the conventional and proposed models, which hinders comprehension for general readers. While the request for clear figures makes perfectly good sense, it is not clear what a Table would consist of. We have not added a Table. The difference between models consists of a major qualitative difference between gating current consisting of large S4 movement and movement of protons. Details may vary, but there is no question as what the difference between models consists of.
Recommended improvements:
- Add a methodology section: Even if theoretical, it should specify which simulations were performed, the level of theory (if applicable), and the rationale behind the model.orce field and the reason for choice of water model, choice of amount of water and lipid to include, and many other details—all of which constitute the rationale for the particular paper. It is impossible to repeat all the methodological sections of all the papers cited. We have discussed what these papers have in common that makes their results questionable. If the simulation methods are all described, the experimental references would have to be included in the methods section. We are only writing a review article, not a complete book.
- Clarify the argumentative structure: Reorganise the manuscript into sections such as classical hypothesis, contradictory evidence, alternative model, theoretical validation, and future projections. I am not sure what kind of paper this would be—I don’t think it makes any sense in this case. For example, one section of the article now discusses the effects of substituting D2O. There is no contrary evidence. The evidence is what it is. T There are multiple simulation papers referred to, each with a force field, and a version of a fhe D2O set of experiments gets its own section. Then it is necessary to move on. In those sections that have possible alternate explanations, such as the “Piquito”, that point is made. The organization of the paper is by category of evidence, with alternate interpretations given in each section. There is a section devoted to the primary evidence that gave rise to the standard model; this section too has its own set of alternate explanations, in this case explanations that are consistent with the proton gating model, together with the original interpretations coming from the standard model. There are experiments, and attempts to explain them. In a sense, the present organization comes as close as is possible to what the reviewer suggests. Future projections are beyond the scope of this article as many groups each add to our knowledge. We cannot guess what every worker in the field will think of.
- Include conceptual figures: For example, comparisons between the S4 displacement model and the H⁺/H₂O/side chain network model. There are several versions of the standard (S4) model, expounded in numerous papers, some of which are cited here. Working through the details of at least most of the proposals in the literature, and commenting on each, is not possible. Here we agree with the reviewer that the figures in our original submission were inadequate. The revised version has new figures, with more detailed labels. However, the new version remedies this.
- Remove unfounded evolutionary speculations (e.g., section 1.20). nThis section is interesting, but not really part of the central argument, and is therefore withdrawn
- The proton-based model is overemphasised without structural validation: the S4 movement is dismissed as artefactual without direct experimental evidence to refute it. This model; this model is not “overemphasized”; it is the subject of the paper. The evidence, section by section, is given in as much detail as seems appropriate. The paper acknowledges that the proton model contradicts what is generally believed, and gives the reasons for reconsidering this general belief.comment, in effect, says the paper should not be written, and the new model should not be considered. Evidence against S4 motion is given section by section—it is what the entire paper consists of. We believe it is time to summarize the evidence that supports the proton gating
- While the criticisms of molecular dynamics (MD) simulations are valid, the manuscript fails to mention recent advances such as polarisable water models or ab initio approaches (e.g., polarisable continuum models, Drude oscillators). In the new version of the paper, this comment is taken into account, with suitable citations; however, the primary criticisms remain, with reasons given.
- The hypothesis of proton tunnelling is not supported by energetic calculations or transition state models. This comment, unfortunately, is true. Energetic calculations are best derived from data on temperature dependence. There is a fair amount of data in the literature giving the Q10 values in the literature. The one thing that is clear Is that the there are several steps, and the overall process is complex. This point is made in the paper, and discussed to the extent present data allows. We agree with the reviewer that understanding the energies involved would be extremely valuable. There is a section in the paper on energetics, but it really comes to no conclusions—it does acknowledge the importance of the topic. We are now doing calculations that will, among other things, partially address this matter, but these calculations are not yet ready for publication.
- The energetic cost of K⁺ dehydration — a key issue in selective transport — is omitted from the analysis. Here the reviewer has a point. We have added some references; the problem is that dehydration energy depends on the environment; the cost of dehydration is well known in vacuum; it is known on average in bulk water. However, it is not so clear in confined water, although we discuss the question to the extent possible. The added references help with this discussion. It is, as the reviewer notes, an important question. However, the fact of dehydration is not in question, even in the standard model; the K+ enters the gate of the channel with six hydrating water molecules, and leaves with at most two, so four have been removed. How this happens, however, is an important question, and we are not entirely ready to answer it. We are doing calculations now ourselves to solve this question, and hope to publish the results later this year. However, for this review article, there is not enough data for a discussion. We do agree with the reviewer that this is an important question.
Comments on the Quality of English Language
The overall quality of English in the manuscript is acceptable but requires significant improvement to meet the standards of an international scientific journal. The text contains numerous grammatical errors, awkward sentence constructions, and inconsistent use of technical terminology. Common issues include incorrect verb tenses, improper punctuation, and unnatural phrasing that can obscure the intended meaning. Additionally, the overuse of conditional language and vague expressions weakens the clarity and strength of the arguments. A thorough revision by a native English speaker or a professional scientific editor is strongly recommended to enhance readability and ensure precise scientific communication.
My (MEG wrote the paper, and this comment is written by him) first reaction to this comment was very negative. I am a native speaker, and I use correct English grammar. However, on rereading the paper, I can understand what the reviewer is saying. There were typos, a number of which were omitted words; the omissions created sentences with grammatical errors. I can see why the reviewer took the English to be inadequate. In addition, I use a number of semicolons, and in small type, these may be mistaken for commas, leading to sentences that appear to be comma-spliced, another error. Hopefully, the paper will appear with the semicolons clear. I am not going to remove them, as they help the prose flow by connecting related thoughts. This said, a major rewrite was called for. The original version was redundant and difficult to read or understand. I have rewritten much of the manuscript, with changes in almost every paragraph. There is some remaining redundancy, as a few questions have to be considered from more than one point of view, so those points are repeated. In addition, in the original version, I left some pronouns with ambiguous antecedents. These have all been disambiguated in the present version. I thank the reviewer for insisting on a rewrite, although I would have picked this up myself on rereading the manuscript. I am not quite sure how I let the original go in poor form, but I do not need an editor to tell me that the first version needed to be rewritten.